# Celiac Disease and Neurological Manifestations: From Gluten to Neuroinflammation

**DOI:** 10.3390/ijms232415564

**Published:** 2022-12-08

**Authors:** Mauro Giuffrè, Silvia Gazzin, Caterina Zoratti, John Paul Llido, Giuseppe Lanza, Claudio Tiribelli, Rita Moretti

**Affiliations:** 1Department of Medical, Surgical and Health Sciences, University of Trieste, 34149 Trieste, Italy; 2The Liver-Brain Unit “Rita Moretti”, Italian Liver Foundation, 34149 Trieste, Italy; 3Department of Life Sciences, University of Trieste, 34128 Trieste, Italy; 4Philippine Council for Healthcare Research and Development, Department of Science and Technology, Bicutan Taguig City 1631, Philippines; 5Department of Surgery and Medical-Surgical Specialties, University of Catania, 95123 Catania, Italy; 6Clinical Neurophysiology Research Unit, Oasi Research Institute-IRCCS, 94018 Troina, Italy

**Keywords:** celiac disease, gluten ataxia, gut microbiota, transglutaminase, transcranial doppler sonography, transcranial magnetic stimulation

## Abstract

Celiac disease (CD) is a complex multi-organ disease with a high prevalence of extra-intestinal involvement, including neurological and psychiatric manifestations, such as cerebellar ataxia, peripheral neuropathy, epilepsy, headache, cognitive impairment, and depression. However, the mechanisms behind the neurological involvement in CD remain controversial. Recent evidence shows these can be related to gluten-mediated pathogenesis, including antibody cross-reaction, deposition of immune-complex, direct neurotoxicity, and in severe cases, vitamins or nutrients deficiency. Here, we have summarized new evidence related to gut microbiota and the so-called “gut-liver-brain axis” involved in CD-related neurological manifestations. Additionally, there has yet to be an agreement on whether serological or neurophysiological findings can effectively early diagnose and properly monitor CD-associated neurological involvement; notably, most of them can revert to normal with a rigorous gluten-free diet. Moving from a molecular level to a symptom-based approach, clinical, serological, and neurophysiology data might help to disentangle the many-faceted interactions between the gut and brain in CD. Eventually, the identification of multimodal biomarkers might help diagnose, monitor, and improve the quality of life of patients with “neuroCD”.

## 1. Introduction

Celiac disease (CD) is an autoimmune disorder triggered by the ingestion of gluten, which, in genetically predisposed people, induces damage to the small intestine and consequent malabsorption [1]. CD has a prevalence of 0.3–1.5% in the general population [2,3]. From a clinical point of view, although diarrhea and gastrointestinal symptoms can commonly be observed at disease onset or in its early phases, they are not as frequent as in the past [1]. It is commonly accepted that the characteristic typical gastrointestinal presentation of the disease represents a small portion of the so-called “CD iceberg”, with more than 50% of adult individuals exhibiting substantial extra-intestinal involvement, even in the absence of the typical CD presentation [1]. Therefore, CD is considered as a complex systemic disorder with multifactorial pathogenesis, arising from environmental exposure to gluten in genetically predisposed individuals.

### 1.1. The History of Celiac Disease

Initially, gluten was identified as the etiological agent by Dicke et al. in 1953 [4]. With the introduction of small bowel biopsies, CD was found to show typical histological features, such as crypt hyperplasia, villous atrophy, and increased intraepithelial lymphocytes [5]. Hence, in the first immunological study published in 1961, circulating antibodies against gluten, called anti-gliadin antibodies (AGA), were identified [6]. Almost a decade later, Marks et al. [7] found similar gut abnormalities in a cohort of individuals with dermatitis herpetiformis, in the absence of any gastrointestinal symptoms. Skin manifestations and gut histological abnormalities were responsive to gluten withdrawal, being this the first demonstration of CD as a systemic pathology.

Up to 20 years ago, CD-related symptoms were tightly bound to malabsorption and vitamin deficiencies. The first patients described with neurological manifestations of gluten sensitivity were malnourished and presented with several vitamins (B1, B6, B12, E) and other micronutrients deficiency [8]. However, more recent clinical findings have shifted CD from a merely malabsorption disorder to an autoimmune disease that affects multiple organ systems, including the nervous system, and that can present with several neurological and neuropsychiatric manifestations.

### 1.2. Pathophysiology of Celiac Disease

Following the ingestion of gluten-containing food, proline-rich gliadin peptides (the antigenic component of gluten) are deamidated by intestinal issue transglutaminases 2 (TG2), a family of enzymes widely expressed in the body [9,10,11], as shown in Figure 1. This results in a gliadin-modified epitope with a greater affinity for human leucocyte antigen (HLA)-DQ2 or HLA-DQ8 pockets in the antigen-presenting cells (APCs). The subsequent CD4 + T-cell activation leads to cytokines release and activation of metalloproteinases (MMP), eventually resulting in mucosal remodeling, intestinal villus atrophy, activation of lymphocytes B, and the production of antibodies to gluten (AGA) and autoantibodies to TG2 (ATG2A) [9,10,11,12,13].

Moreover, studies have demonstrated that gliadin accelerates the disassembling of intercellular junctional proteins via the epidermal growth factor receptor (EGFR) pathway activation [14] and by the downregulation of peroxisome proliferation activated receptor γ (PPAR-γ) gene [15,16,17,18] and intestinal tight junctions (TJ) proteins zonulin-1 (ZO-1), claudin-1, and occluding [15]. The loss of the intestinal barrier leads to increased access for gut-derived molecules (e.g., gliadin antigen itself, biomolecular aggregates, extracellular vesicles, AGA, ATG2A, inflammatory mediators, etc.) to enter the blood circulation and contribute to neurological manifestations by direct toxicity, immune complex deposition, and production of cross-reacting antibodies [16,19], all affecting the so-called “gut microbiota-brain axis” [13,16,20,21,22,23]. Intestinal microbiota-derived products (such as antigens, toxins, miRNA, etc.), indeed, can be found in the brain compartments of CD subjects [14,17].

## 2. Celiac Disease and the Brain

### 2.1. Overview

Neurological symptoms are uncommon in children, whereas approximately 36% of adult patients have neurological symptoms at disease onset [24]. However, the accuracy of these prevalence data is limited because they are usually derived retrospectively from cohorts of gastrointestinal clinics, where patients are unlikely to report neurological symptoms [25]. At the same time, patients presenting with neurological manifestations of CD might not have gastrointestinal symptoms [25].

The most prevalent neurological manifestations in CD are: (1) cerebellar ataxia, also known as “gluten ataxia”, the most frequent neurological disturbance in CD, presenting with progressive gait ataxia, dysphonia, dysarthria, pyramidal signs, and abnormal eye movements [26]; (2) peripheral neuropathy, the second most frequent neurological manifestation of CD (up to 39% of patients), which may even precede gastrointestinal manifestations [27,28,29]; (3) epilepsy, which, in CD, has a prevalence ranging from 3.5% to 7.2% of patients [1,30], although large cohort studies have observed an increased risk of epilepsy in subjects of all ages with diagnosed CD [31]; (4) headache, along with the evidence that in patients with CD a gluten-free diet may reduce the severity and frequency of headache symptoms [32]; (5) mild cognitive symptoms, often referred as “brain fog” (i.e., concentration struggles, episodic memory deficits, world-retrieval difficulties, etc.), which usually improves when gluten-free diet is started and reappears with dietary contamination [33,34]; (6) several psychiatric disorders, including apathy, depression, bipolar disorder, excessive anxiety, irritability, schizophrenia, attention-deficit/hyperactivity disorder, autism, and sleep complaints [35,36,37,38,39].

The causal factors and pathophysiological mechanisms of neurological involvement in CD remain controversial. According to recent evidence, these can be related to a gluten-mediated pathogenesis, including antibody cross-reaction, deposition of immune-complex, direct neurotoxicity, and, in severe cases, vitamins or nutrients deficiency, as shown in Figure 1.

For example, molecular mimicry between the gliadin and proteins of the nervous system has been suggested to play a relevant role in the neurological manifestations of CD [40]. Intestinal TG2 shares high genetic and functional homology with TG6, the brain isoform that is expressed by the activated astrocytes, microglia [41], and neuronal cells [42] of critical brain regions involved in regulating locomotor activity, i.e., basal ganglia, brainstem, cerebellum, globus pallidus, hypothalamus, septal region, some precerebellar nuclei, spinal motor neurons, substantia nigra, and subthalamic nucleus [43]. In CD patients, AGA and ATG6A have shown reactivity with deep cerebellar nuclei brainstem and cortical neurons, resulting in cross-reaction with Purkinje cells and consequent damage [13,29,44,45].

Additionally, common epitopes shared by gliadins and Purkinje cells have been reported [46]. These mechanisms may facilitate barrier breaches in the immune-privileged organs, such as the brain [14,17], although the entire mechanism is still unknown. In *Rhesus* macaques with CD, a high expression of miRNA-204 was found to directly target intestinal TJ protein claudin 1, thus reducing their expression [15]. Interestingly, microglia activation can also initiate the miRNA mechanisms that target genes associated with the innate immune system, TJ, and erosion of glycocalyx at the blood–brain barrier (BBB) [14,47]. A breach in the BBB might also be induced by systemic inflammatory factors (e.g., IL-1, IL-6, IL-8, and TNF-α), such as those produced in the intestine of CD subjects, leading to a persistent low-grade chronic inflammation [48] that, in turn, makes CD patients more sensitive to gluten, even after a gluten-free diet [49].

Finally, the involvement of the blood–cerebrospinal fluid barrier in CD is less studied. Still, an anatomical barrier dysfunction or altered cerebrospinal fluid (CSF) flow rate has been correlated with the presence of IgG anti-dietary antigens (gluten and milk) in the CSF of individuals with first-episode schizophrenia [50].

Lastly, anti-neuronal antibodies may represent an additional etiological factor for neurological involvement in CD. Accordingly, Volta et al. [51] detected the presence of anti-neuronal antibodies in 61% of CD patients with neurological symptoms, with most cases belonging to the IgG class, with titers ranging from 1:50 to 1:200. In more than 50% of patients, the antibodies disappeared after one year of adherence to a gluten-free diet, accompanied by improvement/disappearance of neurological symptoms. The authors also detected anti-neuronal antibodies against the enteric nervous system, without any significant difference in prevalence among patients and controls. Cervio et al. [52] demonstrated in vitro that the exposure of neuronal cells to sera with anti-neuronal antibodies derived from CD patients induced mitochondrial-dependent apoptosis via apoptotic protease activating factor-1 activation through the *Bcl-2* associated X protein and cytochrome c translocation. In a subsequent article, Volta et al. [53] demonstrated the presence of anti-ganglioside antibodies in 64% of patients with CD and neurological symptoms, of whom 50% showed undetectable blood levels after one year of gluten-free diet. Analysis of individual reactive antibody types showed that the anti-GM1 and anti-GD1b subtypes were significantly more frequent in CD patients with neurological dysfunction than in CD patients without neurological symptoms. The authors did not find any correlation between the anti-ganglioside antibody titer and the severity of villous atrophy. However, the triggering factor that determines anti-ganglioside antibody generation remains unknown [54]. Interestingly, a molecular mimicry between microbial antigens, such as lipo-oligosaccharides of *Campylobacter jejuni* and the gangliosides, has been hypothesized as a possible mechanism by which anti-ganglioside antibodies are generated, thus reflecting an abnormal immune response to gut microbial antigens [55,56].

### 2.2. Focus on Gluten Ataxia

In 1966, Cooke et al. [57] described the neurological involvement in a small group of individuals with CD as an inflammatory process involving the cerebellum, eventually leading to the destruction of Purkinje cells. The same inflammation could also affect further regions of the brain, as well as the spinal cord and the peripheral nerves. Later, Hadjivassiliou et al. [58] coined the term “gluten ataxia” to describe the concomitant presence of sporadic idiopathic ataxia and a high prevalence of AGA in the serum. The prevalence of gluten ataxia is about 20–45% (20% when considering all individuals with ataxias, 25% in individuals with sporadic ataxias, and 45% in individuals with idiopathic sporadic ataxias) [59], while the prevalence among AGA-positive patients was 10% in genetically-confirmed ataxias, 10–18% in not genetically-confirmed familial ataxias, and 12% in healthy individuals [60]. The higher prevalence of AGA among idiopathic sporadic ataxias was also confirmed in smaller studies [61,62]. However, these data have not been widely accepted because few studies only achieved an adequate statistical power [63].

Gluten ataxia usually has an insidious onset, with a mean age of presentation of approximately 50 years [64]. Of note, less than 10% of patients with gluten ataxia have gastrointestinal symptoms, although a third show evidence of enteropathy on histology [59]. Gluten ataxia usually presents with pure cerebellar ataxia or, rarely, with ataxia in combination with myoclonus, palatal tremor [59,65], opsoclonus [66], or chorea [67]. In addition, up to 80% of cases of gaze-evoked nystagmus and other visual signs of cerebellar dysfunction have been described, as well as approximately 60% of individuals with combined neurophysiological signs of sensorimotor and axonal neuropathy [59].

Magnetic resonance imaging (MRI) studies showed that up to 60% patients with gluten ataxia have cerebellar atrophy [25] and patchy loss of Purkinje cells across the cerebellar cortex [68]. Using proton magnetic resonance spectroscopy, significant differences in the N-acetyl aspartate concentration at short echo time and N-acetyl aspartate to choline ratios at long echo time in subjects with gluten ataxia, compared to healthy subjects, have been described. These suggest that cerebellar neuronal activity is abnormal in gluten ataxia, although interestingly, the cerebellum was found to be abnormal even in patients without cerebellar atrophy [68].

At the molecular level, ataxia is mainly characterized by the presence of anti-gliadin antibodies related to the HLA haplotype (DQ2, DQ8) [69], anti-Purkinje cell antibodies [60,70], high concentrations of the interferon-γ-inducible chemokine CXCL10 and oligoclonal bands in the CSF [60,70], and the presence of cerebellar inflammation in post mortem investigations [71]. Further evidence supporting immune-mediated pathogenesis involves a widespread infiltration of cerebellar white matter by T lymphocytes, with marked inflammatory perivascular cuffing [10,25]. ATG2A IgA antibodies are detected in approximately 38% of individuals with gluten ataxia. Unlike CD, ATG2A IgG and not IgA antibodies are more prevalent in individuals with gluten ataxia [72]. The high prevalence of IgG class antibodies against TG2 or TG6 is consistent with an immune process involving the central nervous system (CNS) [73]. ATG2A and ATG6A were found in patients affected by idiopathic sporadic ataxia who were negative for AGA [74]. Whether the combined detection of ATG2A and ATG6A IgA/IgG can identify all gluten-sensitive patients remains unclear. However, the discrepancy between ATG2A and ATG6A and AGA detection agrees with the predictable rate of false-positive results and the sensitivity reported for CD [75]. A similar condition exists in the case of gluten ataxia, where the autoimmune response towards the neural anti-transglutaminases might lead to clinical manifestations largely affecting the brain or the peripheral nervous system, with marginal gut involvement [71].

Some studies imply that the autoantibodies related to this manifestation may be an epiphenomenon and irrelevant to the actual pathogenesis, which is related to T cell-mediated mucosal damage [76]. Accordingly, CD may develop in individuals with selective or total antibody-deficiency syndromes, which conflicts with those cases of selective IgA deficiency [71]. IgA deficiency is ten times more common in individuals with CD [77] who still have circulating IgG antibodies [71]. AGA IgG antibodies are particularly relevant in systemic immune response, when compared to IgA class antibodies, which originate in the bowel mucosa [78]. Experimental evidence also suggests the presence of antibody cross-reactivity between antigenic epitopes on Purkinje cells and gluten peptides [79]; additionally, serum from CD patients without neurological symptoms demonstrates cross-reactivity with Purkinje cell epitopes [71].

Nearly all patients with CD have the HLA-DQ2, -DQ8 class II, HLA class II molecules and present antigens derived from exogenous protein to CD4 T cells. As stated before, CD results from an immune response directed towards an exogenous antigen (i.e., gliadin), linked to genetic HLA DQ2/8 expression. Thus, it has been assumed that T cells reactive to gluten peptides have a significant part in disease development in all body districts [71,80]; there is also clear evidence that both humoral and cell-mediated responses are implicated in the pathogenesis of CD and neurological dysfunction [71]. TG2 contributes to ataxia and CD development in at least two ways: by deamidating gluten peptides, increasing their affinity for HLA-DQ2/DQ8, and potentiating the eventual T cell-mediated response [81]. ATG2A are deposited in the small bowel mucosa of CD patients, even in the absence of enteropathy, and in extra-intestinal locations, such as muscle and liver [82]. The broad deposition of ATGA has also been detected in the proximity of brain blood vessels in patients with gluten ataxia [25,63,71]. The deposition was more concentrated in the cerebellum, pons, and medulla. However, it is unclear whether these antibodies originate from systemic circulation or if their synthesis is promoted in the target organs after the stimulation of gut-primed gliadin-reactive CD4+ T cells.

Sárdy et al. [83] detected IgA deposition with TG6 and TG3 around blood vessels in patients with gluten ataxia and dermatitis herpetiformis. These results might imply that the immune complexes originate elsewhere in the body and accumulate around blood vessels, due to an increased vascular permeability, or that they are produced locally by perivascular infiltrating inflammatory cells [25]. TG6 and TG3 can deamidate gluten peptides and synthesize key T cell epitopes. However, there are several differences in IgA sequence specificity affecting the deposition in brain vessels. In particular, perivascular cuffing with inflammatory cells is the result of a vessel-centered process, promoted by perivascular macrophages/dendritic cells in the choroid plexus or the subarachnoid space, determining compromission of the BBB, exposing the CNS to pathogenic autoantibodies, and triggering nervous system involvement. TG2 is normally expressed by brain endothelial cells, especially in the BBB. The binding of ATG2A might initiate an inflammatory response, acting together with additional autoantibodies (e.g., AGA) to cause selective neuronal degeneration, which could result as the consequence of gut ATGA. IgG antibodies are present in approximately 60% of CD patients, whereas the prevalence raises up to 90% in patients with gluten ataxia [59]. Notably, Boscolo et al. [84] showed that patient-derived ATGA can alter, upon exposure, neuronal function in selected areas of the brain, suggesting a mechanism that is independent of the immune system. Additionally, the observed neurological deficits were consistent with the loss of Purkinje cells in patients with ataxia [84]. They also showed a selective reactivity of sera derived from patients with gluten ataxia towards Purkinje cells when applied to the brain tissue.

Nevertheless, although these findings implicate ATGA in ataxia, they were not sufficient to explain the range of distinct neurological deficits that are currently attributed to CD, nor why only a tiny proportion of patients with circulating ATGA antibodies are affected [25]. Antibodies against glutamic acid decarboxylase (GAD), detected in patients with stiff-person syndrome, insulin-dependent diabetes mellitus, and autoimmune polyendocrine syndromes, as well as in some immune-mediated ataxias [85], have been found in 34–60% of individuals with gluten ataxia or in CD patients without neurological symptoms. The introduction of a gluten-free diet can significantly reduce the concentrations and positivity of anti-GAD antibodies (96%) [63]. These observations suggest that the presence of these autoantibodies in enteropathy might promote the development of neurological manifestations [71]. The location-based abundance of GAD within the nervous system (e.g., high concentrations found in Purkinje cells and peripheral nerves) is directly proportional to the severity of the clinical presentation of ataxia and peripheral neuropathy [71,86]. Additionally, GAD in the enteric plexus might explain the possible production of anti-GAD antibodies in patients with CD, thus suggesting that GAD may act as the common antigen linking CD development in the gut to the nervous system [71].

## 3. Celiac Disease, Gut Microbiota, and Inflammation

The gastrointestinal tract is a complex and constantly evolving ecosystem. It is a functional unit, absorbing the primary nutrients essential for human metabolism and preserving the integrity of the mucosal barrier. It also prevents the entrance of pathogens into the body, helping to regulate the immune system function [87]. Therefore, gut microbiota plays a key role both in local and systemic responses in our bodies.

The human microbiota consists of about 10^14^ microorganisms [88], including bacteria, viruses, and yeast, with a gene pool of more than 3 million genes [89] and an estimated weight of up to 1.5–2 kg [90]. As mentioned above, the initial colonization of the gut by microbes begins at birth and is influenced by a large spectrum of perinatal factors. It achieves relative stability around 2–3 years of age [91]. Once established, gut microbiota composition remains almost stable, but some conditions, such as antibiotic treatments, infections, or significant dietary changes, can promote microbial dysbiosis [92]. A “healthy gut microbiota composition” is generally characterized by *Firmicutes* and *Bacteroidetes*, which together represent 90% of gut microbiota, followed by *Proteobacteria*, *Actinobacteria*, *Fusobacteria*, and *Verrucomicrobia* [91,93,94,95]. The *Firmicutes* phylum is divided into more than 200 genera, with the most frequent being the *Clostridium* genera. *Bacteroidetes* consist of Gram-negative and obligate anaerobes, with *Bacteroides* and *Prevotella* being the predominant genera [91,96].

### 3.1. Gut Microbiota and Gluten Digestion

Current evidence suggests that gut microbiota participates in gluten metabolism. *Lactobacilli* and *Bifidobacterium* are involved in the breakdown of gluten, being able to directly metabolize gluten or having extracellular proteolytic activity against gluten metabolites [97,98]. It appears that gut bacteria, both pathogenic and commensal, can produce different breakdown configurations of gluten with increased or decreased immunogenicity, thus influencing the induction of autoimmunity [99]. In particular, *Lactobacilli* can detoxify gliadin peptides after fractional breakdown by human peptidases [99].

### 3.2. Changes in Gut Microbiota in Celiac Disease

In recent years, many studies have investigated salivary, duodenal, and fecal microbiota in patients with CD. These individuals’ gut microbiota is characterized by an increase of potentially pathogenic Gram-negative species (such as *Bacteroides* and *E. coli*) and a decrease in beneficial Gram-positive species (such as *Lactobacillus* and *Bifidobacterium*), compared to healthy subjects [100]. Other studies have reported that Bifidobacterium spp., *Bifidobacterium longum*, *Clostridium histolyticum*, *Clostridium lituseburense*, and *Faecalibacterium prausnitzii* are less abundant in patients with untreated CD [101,102,103].

Ou et al. [104], using scanning electron microscopy, identified rod-shaped bacteria in the epithelial lining of the small intestine of children diagnosed with CD during the so-called “Swedish celiac epidemic”; their presence was not detected in healthy controls [104]. Furthermore, Di Cagno et al. [105] showed that even a 2-years gluten-free diet did not entirely restore healthy microbiota composition in children with CD.

Changes in gut microbiota can interfere with intestinal barrier permeability [106], especially the *Bifidobacteria* strains that have been involved in decreasing epithelial permeability caused by gluten [107], downregulating the Th1 response of CD [108], and reducing the damage of jejunal mucosa architecture [93]. Gut dysbiosis provides intestinal tissue damage, leading to a “leaky gut” status because of intestinal barrier hyperpermeability. Consequently, pathogens can penetrate the bloodstream, promoting the dissemination of inflammation, which can involve different organs and cause extraintestinal manifestations, including neurological and psychiatric symptoms. The gut–brain axis is a bidirectional communication network between the gut and the host nervous system [109], in which data can be exchanged through neural channels, hormones, and the immune system [110]. The brain is an immune-privileged organ, protected by the BBB, that can be damaged by gut dysbiosis via a hormonal network by small molecules, such as lipopolysaccharide (LPS) [111], vascular endothelial growth factors [112], and free radicals [113]. For example, Matisz at al. [114] suggested that gut-related inflammation elicits neuroinflammation by endocrine signals and, therefore, promotes atrophy and hyperactivity in the anterior cingulate cortex, involved in mood disorders, anxiety, and threat-related behaviors [114]. The “leaky gut” condition in CD also encourages the overproduction and release of LPS and long-chain fatty acids, which both play a central role in neuronal plasticity [115,116] and neurodegeneration [117,118,119].

From a molecular point of view, it has been demonstrated that the expression of *PPAR-γ* gene is markedly reduced in CD [15]. *PPAR-γ* is an essential gene with anti-inflammatory and probiotic effects, with its downregulation leading to a change in the microbiota composition, resulting in the increase of *Enterobacteriaceae* (phylum *Proteobacteria*) and reduction in the obligate anaerobic bacteria [120]. Experimental studies conducted on *Rhesus* macaque with CD demonstrated that TJ and their related proteins, such as zonulin, haptoglobin-2, occluding, claudin-1, and zonula occludens-1 (ZO1), were markedly dysregulated in the duodenum of celiac macaques [15]. TJ proteins are known to preserve the integrity of both the intestinal barrier and the BBB, suggesting that the downregulation of the *PPAR-γ* gene and TJ proteins play a pivotal role in raising gut dysbiosis with intestinal inflammation and neuroinflammation [14,15]. Moreover, Mohan et al. [15] described how the “leaky gut” condition promotes the intestinal LPS translocation to the brain and the subsequent activation of microglia, with the development of neuroinflammation via micro-RNA (miRNA) mechanism involving genes associated with the innate immune system and the TJ [15].

In this context, miRNAs are small RNA molecules that regulate gene expression in the post-transcriptional phase and control most cellular processes [15,121]. A significant downregulation of miRNAs (particularly miR-192-5p, miR-31-5p, miR-338-3p, and miR-197) in the duodenal biopsies of celiac patients with severe histopathological lesions has been shown [122]. The downregulation of miR-192-5p was associated with the upregulation of chemokine C-X-C motif ligand 2 (CXCL2) and nucleotide oligomerization domain-2, both involved in innate immune response [122]. Interestingly, miR-195-5p was shown to play a key role in maintaining intestinal homeostasis, since it is downregulated in ulcerative colitis and CD [122,123,124]. Otherwise, in pediatric CD, miR-449a expression was markedly increased and associated with the negative regulation of *Notch* receptor 1 and Kruppel-like factor 4, both involved in the proliferation and differentiation of goblet cells, which are significantly reduced in the small intestines of children with CD [125]. Moreover, increased levels of miR-21 and decreased levels of miR-31 in the serum of CD pediatric patients were considered non-invasive biomarkers of the disease [126]. Finally, miR-21 was associated with the presence of ATG2A IgA antibodies [15,126]. Another study on *Rhesus* macaque with CD confirmed the elevated miR-204 that directly reduces the expression of intestinal TJ protein claudin-1 in the duodenum of celiac macaques [15,19]. Taken together, all the described evidence suggests a potential role of miRNAs in the genesis of intestinal inflammation and the onset of neurological damage in CD patients.

## 4. Biomarkers for Neurological Manifestation of Celiac Disease

Recently, Ramirez-Sanchez et al. [127] provided one of the most comprehensive reviews of the current and future biomarkers for the disease, whose indications are summarized and updated in Table 1. CD is diagnosed by the detection of ATGA-IgA in serum (sensitivity 93%, specificity 95% [10,128]) and, in those cases with high clinical suspicion, by duodenal biopsy [10,128]. Circulating AGA and ATGA may be present in patients with neurological manifestations, even in the absence of intestinal lesions [9]. TG6, expressed in the CNS by activated astrocytes, microglia, and neurons, is considered an autoantigen and an initial marker for neurological involvement in gluten-related diseases, especially in gluten ataxia [41,45]. For this reason, ATG6A has been suggested as a sensitive and specific biomarker in the diagnosis of gluten ataxia [41].

Neurological abnormalities, which are common in CD patients, have also been found in non-CD subjects positive for serological AGA, suggesting alternative causes to gluten exposure for neurologic dysfunction among most AGA–positive patients without CD [129]. Samaroo et al. [130] reported that AGA from schizophrenia patients did not show reactivity to deamidated gliadin peptides. The authors also reported an association between the AGA immune response and ATG2A or HLA-DQ2/DQ8 in CD patients, but not in patients with schizophrenia, proposing that AGA immune response in schizophrenia is independent of the action of transglutaminase enzyme and HLA-DQ2/DQ8. These results indicate that the difference between the antigenic specificity of AGA in individuals with CD and schizophrenia is probably genetically based.

The preferential binding of HLA-DQ2 (90–95%) and HLA-DQ8 (5–10%) to gluten-derived peptides provides a genetic association for CD [131]. However, only 0.4% of CD cases are not HLA-DQ2/DQ8 carriers [132]. Additional genetic predisposition is attributed to an unknown number of non-HLA genes, which encode proteins with an immunological function, such as CTLA-4, IL-2, IL-21, IL-12A, IL-18R1, IL-18RAP, RGS1, and RGS1 [133]. The relevance of genetics to CNS manifestations on CD should be further investigated.

## 5. Neurophysiological Findings in Celiac Disease

As mentioned above, neurological symptoms may be present at the onset of the typical manifestation of CD. Thus, there is a need for the identification of early neurological involvement, progression, and possible complications. In this context, neurophysiological techniques show a central role in allowing the non-invasive assessment of CNS excitability, conductivity, and plasticity.

### 5.1. Electroencephalography

The range of electroencephalography (EEG) abnormalities related to CD is broad, although focal activity (i.e., unilateral or bilateral spikes or slow waves), mostly confined within the occipital regions, have been reported in most of the wakefulness EEG studies in CD patients [134]. However, these findings cannot be considered specific to CD [134]. The frequent occipital lobe involvement in CD is confirmed by increased occipital calcium deposition in these individuals [135,136]. Additionally, the preferential involvement of the occipital lobe may be related to several factors, including its vulnerability to metabolic alterations, such as hypoxia or hypoglycemia, and its thinner morphological structure, compared to the other cortical regions [135,136].

A recent prospective study discovered that individuals with CD were more susceptible to developing epileptiform activities on EEG [135]. However, with strict adherence to a gluten-free diet, a reduction in epileptiform EEG findings was observed [135]. Regarding the impact of alimentary restriction, Parisi et al. [30] found abnormal EEG findings in 48% of the enrolled children with CD [30]. However, these findings were no longer present in most patients after six months of a rigorous gluten free diet.

### 5.2. Multimodal Evoked Potentials

Few studies, most of which used somatosensory evoked potentials (SEPs), have studied patients with CD, detecting altered somatosensory central or spinal dorsal column conduction both in adults and children [137]. However, these findings are not specific to the disease [137]. Additionally, in a large cohort of adult individuals with gluten ataxia, more than 50% had loss/delayed P40 cortical response, indicating dorsal column degeneration [62].

Although CD may impact the auditory system [62], Pawlak-Osinska et al. [138] found no pathological finding in brainstem auditory evoked potentials (BAEPs) and vestibular auditory potentials in a cohort of 30 children with CD, compared to healthy individuals. However, a more recent study by Aksoy et al. [139] reported that 1 of 25 patients with CD had abnormalities in BAEPs, in terms of moderate sensorineural hearing loss.

Patients with CD may develop complications affecting the visual pathways, as evidenced by studies on visual evoked potentials (VEPs) [62], even without direct evidence on neuroimaging [139]. These abnormalities were found to revert after a gluten-free diet [140].

### 5.3. Transcranial Magnetic Simulation

Within common neurophysiological methods, transcranial magnetic stimulation (TMS) is a technique able to non-invasively assess, in “real-time” and in vivo, the functional integrity of the central motor pathway and both intracortical and intercortical excitability, as well as provide insights on the neurochemical changes involved in the different neurological, neuropsychiatric, and systemic disorders causing cognitive impairment [141,142,143]. The assessment of the cortico-spinal conductivity and cortical excitation state is carried out through the application of single- or paired-pulse TMS at an adequate stimulation intensity over the primary motor cortex (M1), thus generating a motor-evoked potential (MEP), which may be easily recorded from specific muscles contralaterally to the stimulation side [144], and that can be repeated over time to monitor disease progression [145].

Globally, TMS-derived evidence in CD suggests a “hyperexcitable celiac brain,” with reduced MEP measures of intracortical inhibition and increased index of cortical facilitation), mainly mediated by GABA and glutamate neurotransmission, respectively [146,147,148]. Recently, the excitation state of interhemispheric connections, mainly due to the transcallosal inhibitory pathway, was tested in de novo CD patients, compared to normal subjects [149]. Transcallosal inhibition resulted to be significantly impaired in CD patients, along with a positive correlation between the degree of disinhibition and cognitive performance, thus further supporting the involvement of GABA-related callosal and cortical circuitry in these patients [149]. Lastly, central cholinergic functioning explored by short-latency afferent inhibition of MEPs was found to be not significantly impaired in newly diagnosed CD subjects, when compared to age-matched healthy individuals, although these patients showed a mild, but significantly worse cognitive performance [150].

## 6. Neuroimaging Findings in Celiac Disease

In the last two decades, several studies have investigated possible neurological changes in imaging in patients affected by CD. For example, Hadjivassiliou et al. demonstrated that, in a cohort of 100 newly diagnosed individuals with CD, 60% had abnormal brain imaging, including abnormal magnetic resonance spectroscopy of the cerebellum (46%) and/or white matter lesions (25%) [25]. In a follow-up study, where the authors included 30 of the original participants investigating the rate of brain volume loss in the cerebellum grey matter, in the thalamus, and in the whole brain, a significantly higher rate of atrophy was found in the cerebellum of those patients who still had autoantibodies positivity on follow-up tests [151].

Lastly, transcranial Doppler sonography (TCD) is an ultrasound technique that, with a ≤ 2 MHz probe transducer, can non-invasively insonate basal cerebral arteries, thanks to a few numbers of narrow skull bone windows. As such, TCD can assess, in vivo, the velocity and resistance of different cerebral blood flow (CBF) vessels, with high temporal resolution and over a sustained period [152,153,154]. Recently, cerebral vasoreactivity and hemodynamics to TCD have been systematically investigated for the first time in a sample of newly diagnosed individuals with CD, who were matched for age and education with healthy subjects [155]. A significant increase in resistivity index and pulsatility index from the basilar artery, following the breath-holding test, was observed in patients, as well as an inverse correlation between these measures and cognitive status [155]. These findings suggested subtle cerebral vasomotor reactivity alterations in the posterior circulation of de novo CD subjects, as well as the degree of involvement of which correlated with worse cognition [155]. Based on the increase of these indices from the basilar artery, it was hypothesized that a subclinical neurovascular involvement in CD may be not only related to clear cerebrovascular accidents, but also to functional or microstructural changes, probably secondary to altered vasoreactivity [156], which appeared to be more evident at the posterior arteries of the brain. This result is in accordance with most EEG studies [134].

## 7. Conclusions

Neurological presentations can affect many patients with CD, to the extent that they may even precede typical CD presentation. In this review, we have highlighted the most important mechanisms, at a molecular level, that may be crucial in the neurological involvement of CD, including the possible role of gut microbiota. Additionally, we reported the significant findings in electrophysiology in CD individuals, which, in most cases, are often subclinical and not specific to CD, although they may revert or progress depending on the adherence to a gluten-free diet. As such, since the gluten-free diet may exert a neuroprotective role, it needs to be introduced as soon as possible, although its effects on neurological manifestations (and particularly on cognitive features) are still debated [157]. Moving from a molecular level to a symptom-based approach, clinical, neurophysiology, and serological data could help to disentangle the many-faceted interactions between the gut and brain in CD. The identification of biomarkers might help diagnose, monitor, and improve the quality of life in patients with CD.

## Figures and Tables

**Figure 1 ijms-23-15564-f001:**
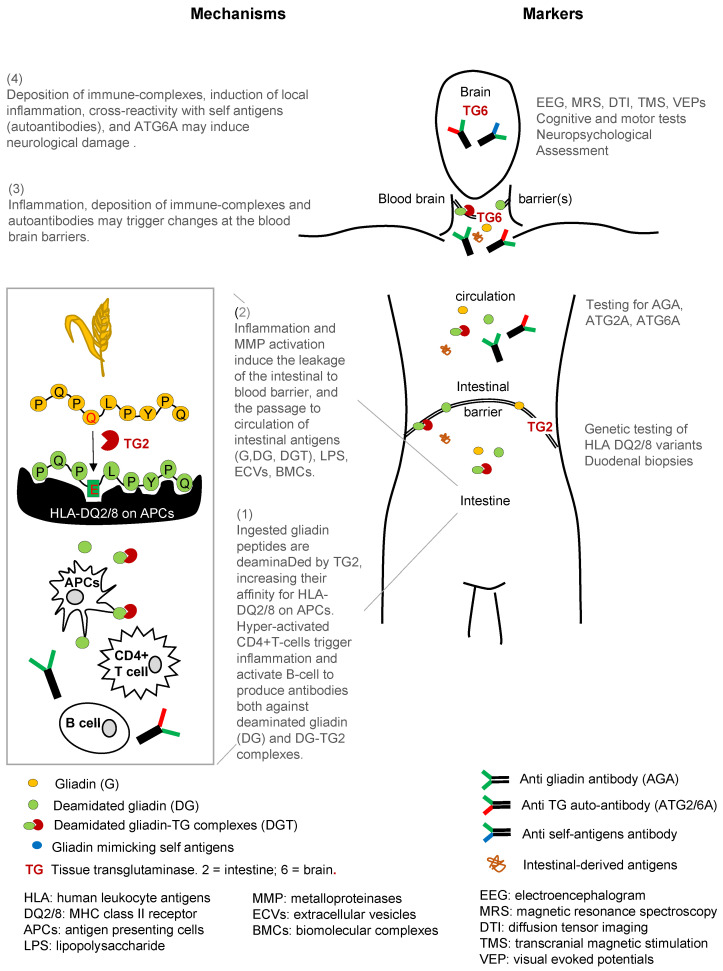
Schematic representation of the proposed pathogenetic changes induced by gluten from the gut to the brain. On the left side (points 1 to 4), the supposed route of action of gliadin derivates, (auto)antibodies, and inflammation mediators from the intestine to the brain. On the right side, the neurological tests and biomarkers of CD. G: in yellow, gliadin (P,Q.P,Q/E,L,P,Y,P,Q; aminoacidic sequence of the gliadin immunogenic epitope), DG: in green, deamidated gliadin, DGT: in green and red, deamidated gliadin-TG complexes, TG: in red, tissue transglutaminase, in blue: gliadin mimicking self-antigens, HLA: Human leukocyte antigens, DQ2/8: MHC class II receptor, APCs: antigen-presenting cell, LPS: lipopolysaccharide, MMP: metalloproteinase, ECVs: extra cellular vesicles, BMCs: bimolecular complexes, AGA: anti-gliadin antibody, ATG2A/6A: Anti TG 2/6auto-antibody, EEG: electroencephalogram, MRS: magnetic resonance spectroscopy, DTI: diffusion tensor imaging, TMS: transcranial magnetic stimulation, VEP: visual evoked potentials.

**Table 1 ijms-23-15564-t001:** Suggested neurological tests and biomarkers of CD. APCs: antigen presenting cell; TG: tissue transglutaminase-gliadin complexes; HLA: Human leukocyte antigens; DQ2/DQ8: MHC class II receptor; ROS: reactive oxygen species; LPS: lipopolysaccharide; TJ: tight junctions; PPAR-γ: peroxisome proliferator-activated receptor gamma; EGFR: epidermal growth factor receptor; BMCs: bimolecular complexes; ECVs: extracellular vesicles; TG2/6: tissue transglutaminase isoform 2 = intestinal/6 = brain; EEG: electroencephalogram, MRS: magnetic resonance spectroscopy; DTI: diffusion tensor imaging, TMS: transcranial magnetic stimulation, VEPs: visual evoked potentials.

Tissue, Organ, or System	Antigen Target	Response	Effects	Biomarker Tool
Intestine	Gluten Gliadin Peptides	TG → gliadin-TG complexes	APCs (high affinity to antigens in HLA-DQ2/DQ8 individuals), Th1 cells.	Genetic testing: HLA-DQ2/DQ8 duodenal biopsy (gold standard)
Intestinal Barrier	Gliadin Gliadin peptides Deamidated gliadin Peptides	APCs, Th2 cells → proinflammatory cytokines and chemokines ROS, LPS; downregulation of TJ & PPAR-γ EGFR pathway activation	APCs, Th1 cells, B cells Antibodies to gliadin Autoantibodies gliadin-TG complex → deposited in tissues	Genetic testing: HLA-DQ2/DQ8 duodenal biopsy intestinal barrier leakage tests
Blood/Systemic Circulation	Gliadin Gliadin peptides Deamidated gliadin Peptides	Proinflammatory cytokines ROS, LPS Biomolecular complexes (BMCs) Extracellular vesicles (ECVs)	Antibodies to gliadin Autoantibodies gliadin-TG complex → deposited in tissues	Genetic Testing: HLA-DQ2/DQ8 Anti-gliadin antibodies Anti-endomysium antibodies Abs: Anti-TGs, Anti-TG2, Anti-TG6
Blood Brain Barrier	Gliadin Gliadin peptides Deamidated gliadin Peptides	Proinflammatory cytokines ROS, LPS Stripping of glycocalyx in endothelial cells → integrity loss	Microglia activation Antibodies to gliadin Autoantibodies gliadin-TG complex Anti-TG6 antibodies	Autoantibodies to gliadin-TG complex
Brain	TG6 expression in CNS structures	Low grade inflammation → increased sensitivity Calcification in occipital lobes	Microglia activation Antibodies to gliadin Autoantibodies gliadin-TG complex Anti-TG6 antibodies	Epileptiform changes, cortical excitability, white matter lesions Using various brain imaging tools: EEG, MRS, DTI, TMS, VEPs
Cerebellar	TG6 expression in cerebellum	Low grade inflammation → increased sensitivity	Anti-TG6 antibodies	Motor dysfunction
Non-Cerebellar	TG6 expression in brainstem, forceps major of corpus callosum, segment of superior longitudinal fasciculus, thalamic white matter	Low grade inflammation → increased sensitivity	Anti-TG6 antibodies	Cognitive tests Neuropsychological assessment

## Data Availability

Not applicable.

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
