# Peer review of "Celiac Disease and Neurological Manifestations: From Gluten to Neuroinflammation"

_ijms, 2022, doi:10.3390/ijms232415564_

Round 1

Reviewer 1 Report

In this review, Giuffrè et al addressed the clinically relevant topic of neurological involvement in celiac disease (CD) patients. Since neurological disorders are not rare in CD patients, an important point is that of biomarkers of neurological manifestations in CD.

-In this regard, the authors should recall and discuss previous important literature data demonstrating the identification of serum autoantibodies significantly related to neurological disorders in CD patients. In particular, it has been previously reported that anti-neuronal antibodies to central/enteric nervous systems antibodies are significantly related to neurological disorders (Clinical findings and anti-neuronal antibodies in coeliac disease with neurological disorders. Scand J Gastroenterol. 2002 Nov;37(11):1276-81). Importantly, antineuronal antibodies might have a pathogenic role via a mitochondrial-dependent apoptosis, as previously demonstrated (Sera of patients with celiac disease and neurologic disorders evoke a mitochondrial-dependent apoptosis in vitro. Gastroenterology. 2007 Jul;133(1):195-206.).

-Other serological markers significantly found in CD patients with different neurological disorders are anti-ganglioside antibodies as previously demonstrated (Anti-ganglioside antibodies in coeliac disease with neurological disorders. Dig Liver Dis. 2006 Mar;38(3):183-7.), and characterized by distinct target autoantigens (Anti-ganglioside antibodies and celiac disease. Allergy Asthma Clin Immunol. 2021 May 28;17(1):53).

Author Response

Dear Editor, we appreciate the opportunity to submit a revised version of the manuscript with reference number “ijms-2042605” to the prestigious International Journal of Molecular Sciences. We are also grateful to the Reviewers for providing their valuable comments and suggestions provided on our manuscript. We have carefully revised the text according to their valuable comments and suggestions (changes are red lined in the revised version). Here is a the point-by-point letter to the Reviewers.

Reviewer #1

Q1: In this review, Giuffrè et al addressed the clinically relevant topic of neurological involvement in celiac disease (CD) patients. Since neurological disorders are not rare in CD patients, an important point is that of biomarkers of neurological manifestations in CD.

-In this regard, the authors should recall and discuss previous important literature data demonstrating the identification of serum autoantibodies significantly related to neurological disorders in CD patients. In particular, it has been previously reported that anti-neuronal antibodies to central/enteric nervous systems antibodies are significantly related to neurological disorders (Clinical findings and anti-neuronal antibodies in coeliac disease with neurological disorders. Scand J Gastroenterol. 2002 Nov;37(11):1276-81). Importantly, antineuronal antibodies might have a pathogenic role via a mitochondrial-dependent apoptosis, as previously demonstrated (Sera of patients with celiac disease and neurologic disorders evoke a mitochondrial-dependent apoptosis in vitro. Gastroenterology. 2007 Jul;133(1):195-206.).

-Other serological markers significantly found in CD patients with different neurological disorders are anti-ganglioside antibodies as previously demonstrated (Anti-ganglioside antibodies in coeliac disease with neurological disorders. Dig Liver Dis. 2006 Mar;38(3):183-7.), and characterized by distinct target autoantigens (Anti-ganglioside antibodies and celiac disease. Allergy Asthma Clin Immunol. 2021 May 28;17(1):53).

A1: We are grateful to the Reviewer for these insightful suggestions. The manuscript has been edited accordingly and a new section that addresses all the comments raised by the Reviewer #1 is now included

Reviewer 2 Report

This paper is a comprehensive and up-to-date review of the complex issue of the neurologic implications found in celiac disease (CD), examined mostly at their pathophysiologic level. I have these additional comments:

I. Table 1 is a bit confusing to me, as it clumps together too many pieces of information, and appears to be possibly erroneous in some respects. For instance: why are APCs listed both under Innate and "Humoral" (adaptive?) immune responses? Also: Transglutaminases gliadin-(tTG) complexes should not be listed under "Innate Immunity". 

II. The AGA (Anti-gliadin antibodies) are repeatedly quoted under various circumstances, but they have been long considered as a very poor marker for CD, as they show poor sensitivity and even poorer specificity. Perhaps a more critical reading of their role should be undertaken.

III. English, though overall correct, shows some areas where it can be improved or must be corrected.
A number of such occurrences are listed:
Physiopathology (row 67) should be instead Pathophysiology.
In figure 1 and in its legend there are several errors: Neuropsycological should be Neuropsychological; deaminated (repeated more than once) should be deamidated; mimiking should be mimicking; Iper-activated should be Hyper-activated.
Also: the expression "gluten pathogenesis" is poor; perhaps better "pathogenetic changes induced by gluten"?
Row 218: "its" should be "their"; row 220: "implicated in autoantibody development, even if we do not know what." --> Very obscure; please clarify.
Row 254: "detected in patients stiff-person syndrome" should be "detected in patients with stiff-person syndrome"

III. Please provide a reference for the rather surprising (to me) statement that peripheral neuropathy is found in up to 50% of CD patients (row 111).

Author Response

Dear Editor, we appreciate the opportunity to submit a revised version of the manuscript with reference number “ijms-2042605” to the prestigious International Journal of Molecular Sciences. We are also grateful to the Reviewers for providing their valuable comments and suggestions provided on our manuscript. We have carefully revised the text according to their valuable comments and suggestions (changes are red lined in the revised version). Here is a the point-by-point letter to the Reviewers.

Reviewer #2

Q1: Table 1 is a bit confusing to me, as it clumps together too many pieces of information, and appears to be possibly erroneous in some respects. For instance: why are APCs listed both under Innate and "Humoral" (adaptive?) immune responses? Also: Transglutaminases gliadin-(tTG) complexes should not be listed under "Innate Immunity"”.

A1: We are grateful to the Reviewer for this comment. We have revised the table accordingly, by adopting an immune response-effect modality.

Q2: The AGA (Anti-gliadin antibodies) are repeatedly quoted under various circumstances, but they have been long considered as a very poor marker for CD, as they show poor sensitivity and even poorer specificity. Perhaps a more critical reading of their role should be undertaken.”

A2: We thank the Reviewer for pointing this out, we agree with this consideration. In the revised manuscript, AGA are discussed in terms of possible molecular mechanism involved in neurological disorders in patients with confirmed celiac disease and not for their role in the disease diagnosis and monitoring, along with relevant references.

Q3: English, though overall correct, shows some areas where it can be improved or must be corrected.
A number of such occurrences are listed:
 Physiopathology (row 67) should be instead Pathophysiology.
In figure 1 and in its legend there are several errors: Neuropsycological should be Neuropsychological; deaminated (repeated more than once) should be deamidated; mimiking should be mimicking; Iper-activated should be Hyper-activated.
 Also: the expression "gluten pathogenesis" is poor; perhaps better "pathogenetic changes induced by gluten"? Row 218: "its" should be "their"; row 220: "implicated in autoantibody development, even if we do not know what." --> Very obscure; please clarify. Row 254: "detected in patients stiff-person syndrome" should be "detected in patients with stiff-person syndrome"”.

A3: Many thanks for that, we have edited the manuscript accordingly.

Q4: “Please provide a reference for the rather surprising (to me) statement that peripheral neuropathy is found in up to 50% of CD patients (row 111).”

A4: Thanks a lot for pointing this out. We have amended the percentage in the text (39%, instead of 50) and included a new reference in the revised version of the manuscript.

Round 2

Reviewer 1 Report

The authors provide a revised and improved version that, in my opinion, can be now accepted.